# Human-animal relationships and interactions during the Covid-19 lockdown phase in the UK: Investigating links with mental health and loneliness

**Elena Ratschen**[1◉]*, **Emily Shoesmith**[1◉], **Lion Shahab**[2◉], **Karine Silva**[3‡], **Dimitra Kale**[2‡], **Paul Toner**[4‡], **Catherine Reeve**[4‡], **Daniel S. Mills**[5‡]

**1** Department of Health Sciences, University of York, York, United Kingdom, **2** Department of Behavioural Science and Health, University College London, London, United Kingdom, **3** Department of Behavioural Sciences, University of Porto, Porto, Portugal, **4** School of Psychology, Queen's University Belfast, Belfast, United Kingdom, **5** School of Life Sciences, University of Lincoln, Lincoln, United Kingdom

◉ These authors contributed equally to this work.
‡ These authors also contributed equally to this work.
* Elena.Ratschen@york.ac.uk

**Data Availability Statement:** All relevant data are uploaded to the OSF repository and publicly accessible via the following URL: https://osf.io/a5uy9/.

## Abstract

### Background

The Covid-19 pandemic raises questions about the role that relationships and interactions between humans and animals play in the context of widespread social distancing and isolation measures. We aimed to investigate links between mental health and loneliness, companion animal ownership, the human-animal bond, and human-animal interactions; and to explore animal owners' perceptions related to the role of their animals during lockdown.

### Methods

A cross-sectional online survey of UK residents over 18 years of age was conducted between April and June 2020. The questionnaire included validated and bespoke items measuring demographics; exposures and outcomes related to mental health, wellbeing and loneliness; the human-animal bond and human-animal interactions.

### Results

Of 5,926 participants, 5,323 (89.8%) had at least one companion animal. Most perceived their animals to be a source of considerable support, but concerns were reported related to various practical aspects of providing care during lockdown. Strength of the human-animal bond did not differ significantly between species. Poorer mental health pre-lockdown was associated with a stronger reported human-animal bond ($b$ = -.014, 95% CI [-.023 - -.005], $p$ = .002). Animal ownership compared with non-ownership was associated with smaller decreases in mental health (b = .267, 95% CI [.079 - .455], $p$ = .005) and smaller increases in loneliness ($b$ = -.302, 95% CI [-.461 - -.144], $p$ = .001) since lockdown.

**Funding:** The authors received no specific funding for this work.

**Competing interests:** The authors have declared that no competing interests exist.

## Conclusion

The human-animal bond is a construct that may be linked to mental health vulnerability in animal owners. Strength of the human-animal bond in terms of emotional closeness or intimacy dimensions appears to be independent of animal species. Animal ownership seemed to mitigate some of the detrimental psychological effects of Covid-19 lockdown. Further targeted investigation of the role of human-animal relationships and interactions for human health, including testing of the social buffering hypothesis and the development of instruments suited for use across animal species, is required.

## Introduction

Human-Animal Interactions (HAI) describe a wide spectrum of interactions and relationships between animals and humans [1] and are of growing interest to researchers, the general public and the media [2]. The ownership of pets (or 'companion animals') and its potential effect on human physical and mental health is a one area of HAI research that has been increasingly popular [3]. Many, mostly observational studies suggest that interactions and relationships with companion animals may be beneficial for human health and wellbeing, for example through hypothesized mechanisms involving attachment to or companionship provided by the animal [3–5]. Despite an increasingly popular belief that living with companion animals may benefit owners, for example in terms of reduced feelings of loneliness and stress through having access to a perceived source of unconditional support, love, comfort, security and stability [6], there is also evidence to the contrary. Some studies show that strong reported bonds with or 'attachment' to companion animals are associated with increased depression and loneliness and can predict vulnerability [5, 6] or increased levels of emotional distress [7] in owners. Studies investigating the link between the human-animal bond and human health are often focused on selected companion animal species, especially dogs and cats, commonly with little or no attention paid to other species. Moreover, these studies tend to be conducted in specific human sub populations (i.e. bereaved individuals; the elderly) [6], and usually differ in the way they conceptualise and measure the human-animal bond. Overall, there is a consensus that substantial scope for further research to explore the likely complex role of HAIs and the human-animal bond on health and wellbeing exists [8, 9]. The potential importance of HAIs that involve non-companion animals, for example farm animals or wildlife, for human health and wellbeing has also been highlighted [10]. This constitutes another area of emerging research in the field.

The Covid-19 pandemic raises previously unexplored questions about the role that human-animal interactions play in the context of widely applied social distancing and isolation measures. Links with human mental health and wellbeing appear of particular interest at this time. Over 40% of United Kingdom (UK) households are estimated to own at least one companion animal [11]. Previously identified psychological effects of infectious disease outbreaks including Covid-19 [12–15], involving stress, anxiety and low mood, render questions related to the potential role of companion animals during this time in the UK especially pertinent. Here we report findings from a survey conducted during the Covid-19 'lockdown' phase in the UK (23 March to 01 June, 2020). During this phase, the government directed people to stay at home except for essential purchases, essential work travel (if remote work was not possible), medical needs, one period of exercise per day (alone or with household members), and providing care for others. We investigated the following questions:

1. What are companion animal owners' perceptions in terms of their animals' roles during the lockdown period, and what concerns do they have in relation to practical aspects of animal ownership during this time? (RQ1)

2. Does strength of the human-animal bond differ by animal species and special role (e.g. assistance dog, emotional support animal) in companion animal owners? (RQ2)

3. What is the association between mental health, wellbeing and reported strength of the human-animal bond in companion animal owners? (RQ3)

4. Are changes in mental health and loneliness scores since lockdown associated with: animal ownership; strength of the human-animal bond; and regular engagement with non-companion animals in non-animal owners? (RQ4)

## Methods

### Study design

Cross-sectional, retrospective survey

### Setting and participants

The survey was conducted in the UK general population. All UK residents over 18 years of age were eligible to take part, irrespective of companion animal ownership.

### Measures

A bespoke questionnaire was generated to enable collection of data most relevant to answer the study questions. It was developed by a multi-disciplinary team of academics with input from third sector animal welfare and training organisations. The questionnaire included validated items and new items based on expert consensus relating to emerging Covid-19-related aspects, as detailed below.

**Demographic data.** Demographic information was gathered about participants' age (in bands, including 70 and above), gender (male/female/non-binary), employment status (e.g. employed, furloughed, retired), level of education (e.g. secondary, Bachelor's degree), housing tenure (e.g. owned or rented), presence of children under 18 years in the household (yes/no), living alone (yes/no), living with partner/spouse (yes/no), and ethnicity (e.g. White, Mixed, Black).

**Covid-19 isolation status.** Information on whether a participant had been in social isolation since the outbreak, e.g. due to a suspected Covid-19 infection was obtained (yes/no).

**Companion animal ownership.** In line with the British Small Animal Veterinary Association (BSAVA), companion animals were defined as 'any domestic-bred or wild-caught animals, permanently living in a community and kept by people for company, enjoyment, work (e.g. support for blind or deaf people, police or military dogs) or psychological support–including, but not limited to dogs, cats, horses, rabbits, ferrets, guinea pigs, reptiles, birds and ornamental fish'). Participants were asked: 'Do you have any animals that live with you or near you, and that you or anyone in your household are the main caretaker of? Please do not include animals kept as livestock (e.g. farm sheep, cattle).' If answering 'yes', they were asked to indicate how many and which species (dog, cat, small mammal, bird, fish, reptile or amphibian, horse or pony, farm animal, other). A question on whether the animal respondents felt closest to was an assistance dog, a therapy animal, an emotional support animal, or another form of working dog was also included; a response option 'none of the above' was provided.

The variable relating to this question was conceptualised as 'animal role' for the purpose of the analyses. Information on whether participants had acquired a new companion animal during lockdown was collected.

Non-animal owners were asked why they did not own animals (e.g. 'I don't like animals'; 'I would like an animal, but my circumstances don't allow it'; 'I have recently lost an animal and am not yet ready to have another one').

**Human-animal bond and interactions.** Participants who owned animals were asked to identify the animal they felt closest to and provide details of the species. With this animal in mind, participants were then asked to indicate agreement to statements on the validated 11-item Comfort from Companion Animals Scale (CCA) [16], using a four-point Likert scale (1 = strongly disagree; 4 = strongly agree). The CCA focuses specifically on the intimacy or comfort domain of the human-animal relationship [16] and is thus more suited to measuring the human-animal bond for a variety of animal species than most other validated instruments, whose 'physical' domain items, for example 'dog walking', are not relevant for other common companion animal species like birds or rabbits [16]. Although the CCA has been commonly referred to in the literature as an instrument to measure 'attachment' to companion animals, we refer to it as an instrument that measures the comfort or 'closeness/intimacy' dimension of the human-animal bond, paraphrased as 'human-animal bond' in the results section for simplicity. Closeness and intimacy are important elements of 'attachment' but do not capture all dimensions described, such as the species-specific behavioural and endocrinological aspects of a deep and unique emotional bond [17]. Scores for each item on the CCA were combined into one total score (11–44) and included as an interval variable in the analyses [18].

Interaction with non-companion animals in all study participants was measured by asking participants whether they engaged regularly in feeding/watching garden birds or other wildlife in their homes or in nature, volunteered or worked for animal rescue organisations or sanctuaries, rode horses that they did not own, visited/watched farm animals near their home, or followed wildlife webcams or Youtube channels and social media groups that regularly share animal videos. Participants were asked whether the time spent on those activities had increased, decreased or remained the same since the lockdown.

**Role of companion animals during lockdown.** Participants who owned animals were asked to identify the animal they felt closest to, and indicate their agreement with seven statements on the role of their animals in the lockdown situation on a 4-point Likert scale (1 = strongly agree; 4 = strongly disagree): 'my animal helps me cope emotionally with the Covid-19 situation'; 'my animal keeps me fit and active in the Covid-19 situation'; 'my animal is the reason I keep in touch with some people or social media groups'; 'my animal has positive effects on my family at this time'; 'my animal causes problems in my family at this time'; 'I can't imagine being without my animal at this time', and 'It would be easier for me not to have an animal at this time'.

Participants were also asked whether they felt worried about their animals because of Covid-19 and if so, to indicate the reason among nine pre-specified options, or to specify it themselves. The pre-specified reasons included: financial difficulties; complications buying pet food; restrictions to veterinary treatment and care; not knowing who would look after the animal if the owner fell ill; restrictions to walks/exercise; possibility of getting infected with Covid-19 by the animal; the animal not coping well when owner returns to work; the animal not coping well with changes in household routine. A question on whether participants had considered or were considering relinquishing their animal was included.

**Mental health, wellbeing and loneliness.** The Short Warwick Edinburgh Mental Wellbeing Scale (SWEMWBS) [19], the mental health subscale of the SF-36 (MHI-5) [20], and the 3-item short version of the UCLA loneliness scale [21] were included. Higher scores on these scales represent better wellbeing, mental health and greater loneliness, respectively. The MHI-

5 scale and the 3-item loneliness scale were used to collect current and retrospective data, asking participants to indicate their perceptions for the time 'before lockdown' and the present time at questionnaire completion (during lockdown), respectively.

**Remote social contact during Covid 19.** Frequency of remote contacts with non-cohabitating family members and friends was measured, with responses 'many times per day'; 'at least once per day'; 'at least every other day'; 'at least twice per week'; 'at least once per week'; 'less than once per week', and 'I do not have any family members or friends I am in regular contact with'.

## Recruitment and procedures

The survey was released in Qualtrics survey software and promoted using academic and third sector networks including animal charities with an interest in human-animal interaction research, social media (Facebook, Twitter) and other media outlets (e.g. Reddit). Prospective participants followed a link to the survey where they were presented with a Participant Information Sheet and consent form. Consent to participate in the anonymous survey was indicated by ticking an online check box. The Participant Information Sheet included an overview of the study and its aim to investigate the role human-animal relationships and interactions play in the context of the Covid-19 pandemic. A screening question requiring participants to name their country of residence denied access to non-UK residents. All data were stored on the Qualtrics server at the University of York.

The study commenced on 16 April 2020, four weeks after strict social distancing and social isolation measures came into force in the UK, and ended on 31 May, when lockdown measures were officially eased, allowing for more extensive travel and gradual relaxing of social distancing rules.

Ethics approval for the survey was granted by the Health Sciences Research Ethics Committee at the University of York, UK, on 16 April 2020.

## Data analysis

Descriptive summary statistics are presented for demographic variables and data relating to animal ownership, species, animal owners' worries related to animal ownership, and animal owners' perceptions about the role of their animals during the Covid-19 pandemic (RQ1).

To investigate whether the strength of the human-animal bond differed by animal species and/ or special role (e.g. assistance dog) in companion animal owners (RQ 2), we conducted separate ANOVAs to assess the reported level of the human-animal bond (measured by the continuous total score of the CCA) by species and by role (e.g. assistance dog), respectively. In addition, a separate multivariable linear regression was constructed to look at the association between animal species and role with human-animal bond, adjusting for age, gender, Covid-19 social isolation status, loneliness (UCLA score-present), mental health (MHI-5 score-present), and living with children.

To explore the association between mental health, wellbeing and reported strength of the human-animal bond in animal owners (RQ 3), we conducted a linear regression to assess the association between human-animal bond and mental health score pre-lockdown (at 'baseline'), acknowledging that human bonds with companion animals have been shown to be stable over time [22], controlling for loneliness (pre-lockdown), gender, age, living with partner/spouse and species. We also conducted linear regression analyses to assess whether the human-animal bond was associated with the outcome variables mental health (present) and wellbeing (present), adjusting for the same covariates (but including loneliness at present).

To understand whether changes in mental health and loneliness scores since lockdown were associated with animal ownership; strength of the human-animal bond; and regular engagement with non-companion animals in non-animal owners (RQ 4), separate linear regression analyses were conducted. These assessed the association between each predictor (animal ownership: yes/

no; animal species; human-animal bond (continuous score); living alone: yes/no; social contacts: none and less than once a week/at least once a week; regular engagement with non-companion animals: none/a minimum of one activity; Covid-19 isolation status: yes/no), and the change scores for loneliness and mental health since lockdown (outcome variables), adjusting for relevant covariates (gender, age, living with partner/spouse; and ethnicity and housing tenure for the comparison between animal owners and non-animal owners). Throughout all analyses, missing values on covariates were imputed using an imputation model with all other variables as predictors: ten imputed datasets were created, each analysed separately, and the results were combined to produce pooled estimates of effects; allowing the analyses to account for uncertainty caused by estimating missing data. Pooled estimates are reported throughout.

Data were analysed using SPSS version 26.0 (IBM®). Standard alpha-levels were applied in two-tailed tests of significance ($p < .05$ considered significant), with family-wise error rate corrected using the false discovery rate [23]. All analyses were pre-specified and uploaded on the Open Science Framework (https://osf.io/jkt9y/).

## Results

A total of 6,007 eligible participants consented to taking part in the study. Of those, 81 had answered no more than the first two survey questions and were removed for the purposes of data imputation and analysis, resulting in a final sample of 5,926 participants. A large majority (89.7%; n = 5,391) had completed the questionnaire in full, without any missing data. A summary of participant characteristics is presented in Table 1; complete participant characteristics are presented in the S1 Table.

A large majority of the sample were companion animal owners (89.8%; n = 5,323), of whom 4.6% (n = 246) reported having added an animal to their households during the Covid-19 pandemic. The most common companion animal species owned were dogs and cats (69.9%; n = 3,719 and 44%; n = 2,340, respectively), followed by small mammals (9.8%; n = 519), fish (9.1%; n = 485), horses or ponies (6.3%; n = 334), birds (5.3%; n = 282), reptiles (3.9%; n = 208), farm animals (1.2%; n = 65) and amphibians (0.7%; n = 37). Having one companion was most common (41%; n = 2,183); 24.2% (n = 1289) of participants had two, 24% (n = 1,276) between three and six, and 10.8% (n = 575) had seven animals or more. Companion animals with special roles were owned by 9% (n = 481) of participants: there were 4.7% (n = 251) emotional support animals, 2.3% (n = 123) therapy animals, 1.1% (n = 57) assistance dogs (e.g. guide dogs), and 0.9% (n = 50) working dogs reported in our sample.

Of participants who did not own animals (10.2%, n = 603), 48.6% (n = 293) reported that they would like to have an animal but that current circumstances did not allow it, 18.9% (n = 114) reported they liked animals, but did not want to have one, and 16.4% (n = 99) said they had been planning to get an animal for some time; 13.6% (n = 82) reported having recently lost an animal and not being ready to have another one yet; and 2.5% (n = 15) reported they did not like animals.

Just over a fifth (21.1%; n = 127) reported they were considering to acquire or foster an animal due to the Covid-19 situation.

### What are companion animal owners' perceptions in terms of their animals' roles during the lockdown period, and what concerns do they have in relation to practical aspects of animal ownership during this time? (RQ1)

The vast majority of companion animal owners stated that their animals constituted an important source of emotional support (Fig 1), with dogs, cats, horses and other companion farm animals scoring highest across most support domains (Table 2).

**Table 1. Participant characteristics.**

| Characteristics | | % (N) |
|---|---|---|
| **Gender** | Female | 78.6 (4,657) |
| | Male | 20.6 (1,222) |
| | Other | 0.6 (36) |
| | Prefer not to say | 0.2 (11) |
| **Age (years)** | 18–24 | 7.1 (420) |
| | 25–34 | 17.5 (1,040) |
| | 35–44 | 16.8 (994) |
| | 45–54 | 23.8 (1,409) |
| | 55–64 | 22.2 (1,313) |
| | 65–70 | 7.1 (418) |
| | Over 70 | 5.6 (332) |
| **Ethnicity** | White | 96.9 (5,742) |
| | Mixed/multiple ethnic | 1.1 (67) |
| | Asian/Asian British | 0.5 (32) |
| | Black/African/Caribbean/Black British | 0.1 (6) |
| | Chinese | 0.1 (8) |
| | Arab | 0.1 (4) |
| | Other ethnic | 0.3 (15) |
| | Prefer not to say | 0.9 (52) |
| **Companion animal ownership** | Yes | 89.8 (5,323) |
| **Interaction with non-companion animals** | Feeding/watching birds in my garden | 54.5 (3,228) |
| | Feeding/watching other wildlife in my garden | 28.4 (1,685) |
| | Watching wildlife in nature | 39.5 (2,346) |
| | Volunteering/working for animal rescue organisations or sanctuaries | 5.6 (330) |
| | Riding horses that I am not the main caretaker of | 1.6 (97) |
| | Visiting/watching farm animals close to my home | 13.9 (822) |
| | Following wildlife webcams online | 11.8 (698) |
| | Other enjoyable activities involving animals that you are not the main caretaker of | 8.1 (479) |
| | None of the above | 24.8 (1,468) |
| | Other | 8.7 (515) |
| **Cohabitation** | Live alone | 18.2 (1,081) |
| | With partner/spouse | 61.3 (3,630) |
| | With children < 18 years | 21.1 (1,250) |
| | With adults 18–70 years old | 22.8 (1,349) |
| | With adults > 70 years | 3.1 (184) |
| | With persons who may be vulnerable to Covid-19 | 9.8 (579) |
| **Covid-19 social isolation status** | Socially isolating | 37.4 (2,219) |
| | Not socially isolating | 62.6 (3,707) |

The majority of companion animal owners (67.6%; n = 3,599) reported having been worried about their animal(s) because of Covid-19, most frequently due to restricted access to veterinary care (40.8%; n = 2,171), because they wouldn't know who would look after the animal (s) if they fell ill (22%; n = 1,173), because of restrictions to exercise/walks (21.7%; n = 1,155), because of concerns that the animal(s) wouldn't cope well when participants' returned to work after the pandemic (19%; n = 1,012), and because obtaining pet food had become complicated (17.8%; n = 948). Nearly all (99.7%; n = 5,307) reported they had not considered giving up their animal(s) since the start of the pandemic.

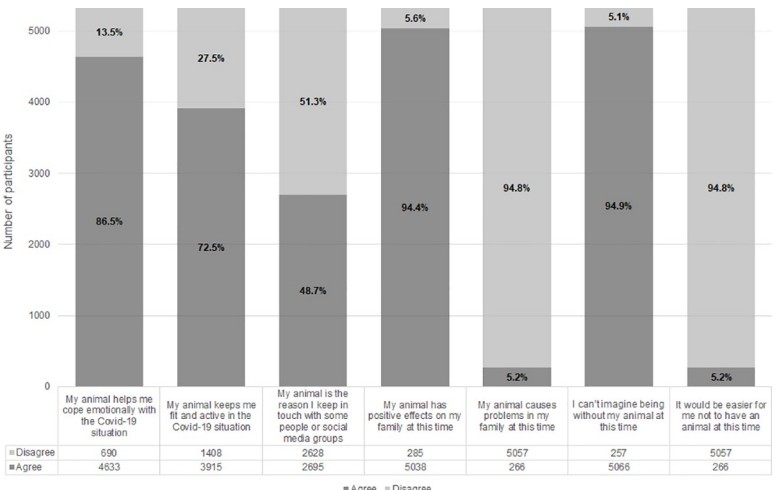

**Fig 1. Perceptions of owners regarding the role of companion animals during Covid-19.**

## Does strength of the human-animal bond differ by animal species and special role in companion animal owners? (RQ2)

ANOVA analyses showed there was a significant association between the human-animal bond and animal species ($F(9, 5,196) = 5.04$, $p < .001$) but not with animal role. Post hoc tests showed that participants with small mammals, birds, fish and reptiles had a significantly lower CCA score than participants with dogs and horses, while participants with fish and reptiles had a significantly lower CCA score than participants with cats ($p < .05$; see Fig 2).

The multivariable linear regression showed that animal species was significantly associated with human-animal bond even when controlling for all covariates ($b = -.198$, 95% CI [$-.341$ $-.055$], $p < .05$), but became non-significant when animal role was included ($b = -.059$, 95% CI [.688 $-.570$], $p > .05$).

## What is the association between mental health, wellbeing and reported strength of the human-animal bond in companion animal owners? (RQ3)

Adjusting for relevant covariates, higher CCA scores were significantly associated with lower mental health scores pre-lockdown. However, they were not significantly associated with mental health and wellbeing scores since lockdown, although higher CCA scores were approaching significance with lower mental health scores since lockdown (see Table 3).

**Table 2. Perceptions of owners regarding the role of companion animals during Covid-19, grouped by animal species.**

|  | Dogs | Cats | Horses, ponies and farm animals | Other |
|---|---|---|---|---|
|  | Agree % (N) | Agree % (N) | Agree % (N) | Agree % (N) |
| My animal helps me cope emotionally with the Covid-19 situation. | 91.2 (3,106) | 89.3 (1,272) | 94.6 (88) | 86.5 (167) |
| My animal keeps me fit and active in the Covid-19 situation | 96.4 (3,285) | 31.8 (453) | 95.7 (89) | 45.6 (88) |
| My animal is the reason I keep in touch with some people or social media groups | 59.1 (2,013) | 35.1 (501) | 86.0 (80) | 52.3 (101) |
| My animal has positive effects on my family at this time | 98.9 (3,371) | 98.3 (1,401) | 89.2 (83) | 94.8 (183) |
| My animal causes problems in my family at this time | 5.1 (174) | 4.6 (66) | 10.8 (10) | 8.2 (16) |
| I can't imagine being without my animal at this time | 99.6 (3,392) | 98.4 (1,402) | 99.0 (92) | 93.3 (180) |
| It would be easier for me not to have an animal at this time | 5.0 (172) | 4.3 (61) | 18.3 (17) | 8.2 (16) |

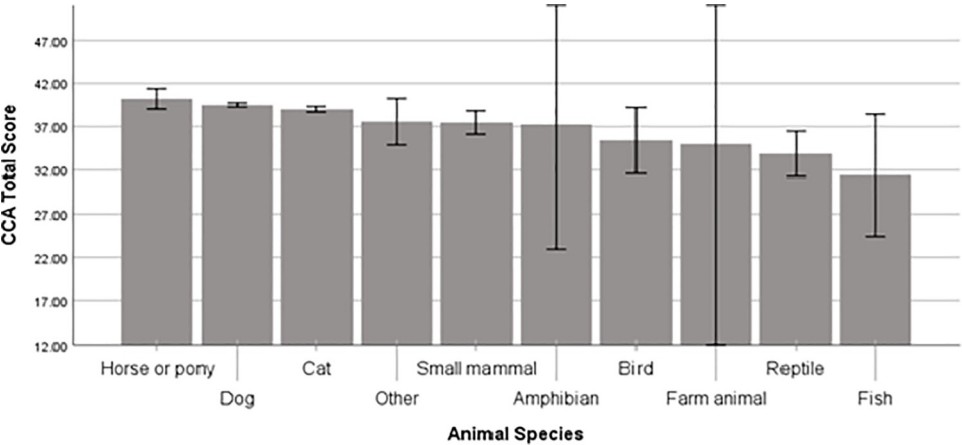

**Fig 2. CCA total score grouped by animal species.**

## Are changes in mental health and loneliness scores since lockdown associated with: Animal ownership; strength of the human-animal bond; and regular engagement with non-companion animals in non-animal owners? (RQ4)

Adjusting for relevant covariates, animal ownership was significantly associated with smaller decreases in mental health scores and smaller increases in loneliness scores, as indicated by MHI-5 and UCLA change scores. Mental health and loneliness scores for animal owners and non-owners, pre- and since lockdown, are provided in the S2 Table. Living alone and higher frequency of reported remote social contact were significantly associated with higher increases in loneliness scores, but not with mental health change scores. The remaining predictors were not significantly associated with the loneliness or mental health change scores (see Tables 4 and 5).

## Discussion

To the best of our knowledge, this is the first study to investigate aspects related to companion animal ownership, the human-animal bond and mental health in a sample of this size, and the

**Table 3. Linear regression models of association of CCA scores and mental health and wellbeing scores, adjusting for relevant covariates.**

| Predictor | Mental health (pre-lockdown) | | | |
|---|---|---|---|---|
| | $b_{adj}$ | 95% CI | *p*-value | $R^2$ |
| CCA scores[1] | -.014 | -.023 –-.005 | .002* | .106 |
| | Mental health (since lockdown) | | | |
| | $b_{adj}$ | 95% CI | *p*-value | $R^2$ |
| CCA scores[2] | -.009 | -.017 –.001 | .051 | .153 |
| | Wellbeing (since lockdown) | | | |
| | $b_{adj}$ | 95% CI | *p*-value | $R^2$ |
| CCA scores[3] | .010 | -.006 –.026 | .225 | .244 |

[1]Gender, age*, living with partner/spouse*, animal species, loneliness score (pre-lockdown)*

[2]Gender*, age*, living with partner/spouse, animal species, loneliness score (since lockdown)*

[3]Gender*, age*, living with partner/spouse*, animal species, loneliness score (since lockdown)*

**Table 4. Linear regression models of association of predictors and the change score for loneliness, adjusting for relevant covariates.**

| Predictor | Loneliness (change score) | | | |
|---|---|---|---|---|
| | $b_{adj}$ | 95% CI | *p*-value | $R^2$ |
| **Animal ownership**[1] | -.302 | -.461 --.144 | .001* | .027 |
| **CCA scores**[2] | .002 | -.005 -.009 | .599 | .024 |
| **Animal species**[3] | -.012 | -.048 -.025 | .537 | .024 |
| **Engagement with non-companion animals**[4] | -.213 | -.644 -.218 | .333 | .020 |
| **Living alone**[5] | .611 | .490 -.731 | .001* | .020 |
| **Social contact**[6] | .424 | .222 -.626 | .001* | .023 |
| **Isolation status**[7] | -.014 | -.109 -.082 | .777 | .021 |

Gender*, age*, ethnicity*, housing tenure*, living with partner/spouse*

[2] Gender*, age*, living with partner/spouse*

[3] Gender*, age*, living with partner/spouse*

[4] Gender, age, living with partner/spouse

[5] Gender, age

[6] Gender, age*, living with partner/spouse*

[7] Gender*, age*, living with partner/spouse*

first substantial survey to collect detailed information on special companion animal roles (e.g. assistance dogs; emotional support animals), and non-companion animal-related human-animal interaction in this context.

Results from this survey suggest that companion animals constituted an important source of emotional support to owners in the Covid-19 lockdown, with no statistically significant differences in the emotional/intimacy dimension of the human-animal bond identified between animal species in the fully adjusted model. Interestingly, stronger reported human-animal bonds were associated with poorer mental health pre-lock down, highlighting that close bonds with animals may indicate psychological vulnerability in owners. However, having a

**Table 5. Linear regression models of association of predictors and the change score for mental health, adjusting for relevant covariates.**

| Predictor | Mental health (change score) | | | |
|---|---|---|---|---|
| | $b_{adj}$ | 95% CI | *p*-value | $R^2$ |
| **Animal ownership**[1] | .267 | .079 -.455 | .005* | .009 |
| **CCA scores**[2] | .006 | -.003 -.014 | .190 | .007 |
| **Animal species**[3] | -.021 | -.065 -.023 | .346 | .006 |
| **Engagement with non-companion animals**[4] | -.208 | -.677 -.260 | .384 | .015 |
| **Living alone**[5] | .079 | -.064 -.222 | .277 | .007 |
| **Social contact**[6] | -.035 | -.275 -.204 | .771 | .007 |
| **Isolation status**[7] | -.082 | -.195 -.031 | .157 | .007 |

Gender*, age*, ethnicity*, housing tenure, living with partner/spouse

[2] Gender*, age*, living with partner/spouse

[3] Gender*, age*, living with partner/spouse

[4] Gender, age*, living with partner/spouse

[5] Gender*, age*

[6] Gender*, age*, living with partner/spouse

[7] Gender*, age*, living with partner/spouse

companion animal, but not strength of the human-animal bond, was associated with less deterioration in mental health and smaller increases in loneliness since lockdown. This suggests that aspects of non-specific social support associated with ownership may make owners more resilient in the context of the lockdown.

Generalisability of our findings is limited by several factors. Firstly, the study population was a convenience sample that is not representative of the UK general population, as it consisted largely of female companion animal owners. Such bias is common in the field of HAI research [24]. Although gender differences have been identified for some aspects relating to human-animal interaction [25], this does not seem to be the case for certain facets linked to the intimacy domain of the human-animal bond (e.g. self-disclosure [26]). Thus, our sample may not have affected our results substantially. Also to be noticed is the fact that the majority of participants who did not own an animal reported that they would like to or were planning to have one. Hence, this survey was evidently a 'survey of animal lovers'. Results need to be interpreted with this caveat in mind–especially where statistical comparisons between animal owners and non-owners are concerned [24]. Moreover, the population of non-owners in this survey may not be representative of non-owners in general; just 2% said they did not like animals, when the national average is thought to be around 10% [27].

A further limitation refers to our questionnaire instrument. Due to the particularities of the Covid-19 lockdown situation and social distancing requirements, we devised a new short item pertaining to frequency of remote social contacts, to complement the validated short UCLA loneliness measure, rather than depend on standard validated instruments on social support developed for other contexts (e.g. PRQ85 [28] and Close Persons Questionnaire [29]). We accept that the 'remote social contact' variable is not validated and, even in conjunction with the UCLA measure, may not cover the social support dimension as comprehensively as other instruments. However, our results suggest that the potential role of social support may be a particularly important area for future research.

## The human-animal bond as potential indicator of psychological vulnerability in companion animal owners

While it is commonly assumed that companion animals can be beneficial for the health and welfare of most owners, the quality of research and bias towards reporting favourable results has led some to challenge this perspective [30]. Contradictory views might arise, at least in part, from different study population characteristics: it is notable that many studies investigating physical benefits of companion animal ownership, including one which is widely considered to be the scientific pioneer in this regard [31], have focused on the effect of companion animals on individuals who were already sick when the research commenced. Likewise, many of the studies on psychological benefits have focused on participants with existing mental health problems [3]. Importantly, a relatively substantial Swedish epidemiological study [32] found that companion animal owners suffered more psychological problems than non-owners, and some smaller HAI-focused studies [5, 6] also indicate that strong or close relationships with animals might predict owners' mental health vulnerability. This aligns closely with our own results, which showed that lower mental health scores at baseline were significantly associated with higher reported human-animal bond scores in adjusted analysis, potentially indicating mental health vulnerability. We did not identify significant associations between strength of the human-animal bond and mental health or wellbeing since lockdown. This is perhaps unsurprising, given the general deterioration in mental health and wellbeing scores during lockdown found in our own and in other recent studies [33]. Human-animal bonds have been described as authentic and stable emotional relationships [4, 22], and the significant

association we identified between strength of bond and mental health score at baseline appears to reflect this. Further exploration of the potential of appropriate human-animal bond measures as possible adjunct screening instruments for mental health risks may be warranted. Asking animal owners about closeness to their animals to identify potential mental health vulnerabilities could potentially offer advantages in certain clinical contexts, for example in sub populations with whom direct conversations about mental health or administration of mental health scoring instruments may be challenging. Furthermore, seeing as those with a stronger bond to their animals might be more likely to engage in self-selecting HAI studies, such as the one reported here, it seems important for longitudinal and interventional studies in the area to evaluate the strength of the bond and mental health status of participants at baseline to uncover potential bias.

### The role of the bond versus the presence of the individual: Why 'species' may not really matter

Although univariate analyses demonstrated that horses, dogs and cats scored significantly higher in terms of the CCA score, fully adjusted multivariate adjusted analysis that included animal role as a covariate did not. Emotional closeness perceived by humans in relation to a companion animal, therefore, does not appear to depend significantly on animal species. It is notable how variable strength of the bond was in relation to some types of animal, which one may not typically associate with emotionally close relationships (e.g. amphibians). This is an important finding in light of discourses that tend to emphasize the differences in human-animal bond by animal species [16, 34]. It is often assumed that the human-dog relationship occupies a special status with regard to impact on human health animals [35] above and beyond the relationship with other animals. However, a more recent critical perspective suggests that it may be the adaptability of the dog to working in different ways and possibly its sensitivity to human emotional cues [36], which may primarily underpin its widespread use in animal assisted interventions, rather than the interpersonal relationship. Our finding that the strength of the human-animal bond did not vary significantly by species might support the social buffering hypothesis [37]: the presence and recognition of an animal of any kind as part of the human's social group may be more important for shaping the relationship than species-specific aspects.

Given the complex and dynamic interaction between the human-animal bond and human related factors [38], there is currently little consensus on the terminology used to describe or quantify human-animal relationships [39]. The instruments available are generally limited, especially for measuring the human-animal bond in adults, and often focus on the human-dog relationship [39]. Most, including the CCA used in this study, propose some form of 'attachment' as their conceptual basis, but this term is itself inconsistently defined. While used very broadly by some to describe the relationship underpinning many aspects of prosocial behaviour [40], 'attachment', as first developed by Bowlby to describe human-human relationships [41], refers to the specific bond that develops between a dependent and a care-giver, but not the converse. Characterising and measuring it would require capturing a variety of attachment-specific aspects, including, for example, specific behaviours related to the maintenance and re-establishment of proximity, and endocrinological responses (e.g. involving oxytocin) [17]. We question whether focusing on 'attachment' continues to be useful or necessary to conceptualise the human-animal bond and associated human-animal relationships. The concept of social buffering [37, 42], whereby affiliated social partners within a social group can mitigate stress responses and increase resilience in individuals, may be an important starting point for future research. Such research could attempt to integrate both positive (e.g. human-

animal bond) and negative (e.g. caring responsibilities) aspects of human-animal relationships. Both aspects were reported to play a substantial role for participants in our study context.

## Caring for companion animals during lockdown: Emotional support and concerns

The vast majority of animal owners perceived their animals to help them cope with the pandemic context and reported that they constituted an important source of emotional support. However, concerns and worries relating to caring adequately for animals at this time, when access to, for example, veterinary care, animal feed and adequate outdoor exercise spaces was limited, were also frequently reported. In the context of the continuing pandemic that is likely to affect society for some time, it will be important to understand how Covid-19-specific concerns and worries related to animal ownership may affect different subgroups of the population, especially the elderly and those shielding or socially isolating. Many animal charities, veterinary care providers and other organisations in the field have already developed first sets of guidelines and resources for owners, providing support with handling common challenges (e.g. the British Small Animal Veterinary Association: https://www.bsava.com/adviceforpetowners). Building on these efforts by developing further resources and identifying multi-sector and multi-agency support options appears to be important. In light of widespread concerns that the Covid-19 pandemic may continue to bring with it a surge in the abandonment of companion animals [43], this may be particularly pertinent. Even though hardly any of our study participants (0.2%) stated that they had considered giving up their animals during the pandemic, reporting bias, especially in terms of social desirability [44], related to this particular question may have been high, and, as previously discussed, our sample was not necessarily representative of the UK general companion animal owning population.

Just over 2% of participants in our sample reported having permanently acquired a new animal since the lockdown 'due to the Covid-19 situation', and 0.8% had started to foster a rescue animal. We did not collect data on whether these participants were experienced animal owners, what the reasons for acquiring or fostering an animal were, and whether the acquired or fostered pets were added to households that already owned pets. Longitudinal studies investigating aspects of animal welfare and behaviour in the context of the Covid-19 pandemic are underway; a cross-sectional study from Spain found that cats and dogs demonstrated signs of behavioural change consistent with stress during lockdown [45]. Further research investigating behavioural change across companion animal species seems indicated. It would be important to understand whether presumed anticipated mental health benefits for humans may play a part in 'animal acquisitions' during lockdown, especially in view of our findings discussed below.

## Animal ownership and its links with changes in mental health and loneliness since lockdown

In our study population, having a companion animal was associated with decreased deterioration in mental health and smaller increases in loneliness since lockdown. Importantly, mental health deteriorated and loneliness increased for both animal owners and non-owners after lockdown, and although statistically significant, the difference in changes between animal owners and non-owners was very small (0.28 and -.029 mean score difference for loneliness and mental health, respectively)–most probably not of clinical importance, if extrapolating the concept of the minimally clinically important difference (MCID) [46] to our study context. However, our study sample was not clinical, and it is possible that mitigating effects of animal ownership may have differed in a clinical context, with more detailed validated instruments to

measure mental health-related outcomes applied. Overall, our finding does not justify any potential assumption that a companion animal in general strongly protects against worsened mental health and increased loneliness in a pandemic context. With the promotion of the 'mental health benefits of pets' increasingly prolific and recently including claims related to immediate benefits during the pandemic (e.g. https://www.helpguide.org/articles/healthy-living/how-pets-can-help-you-cope-during-coronavirus.htm), this is an important consideration. Nonetheless, in our sample of 'animal lovers', having an animal was linked to somewhat attenuated effects of the lockdown experience on mental health and loneliness. This finding resounds with the wider literature relating to the positive effect companion animals can have on human mental health, especially in terms of feelings of loneliness, through various postulated mechanisms [8, 47], which we believe should include social buffering. The latter may explain why changes in mental health and loneliness since lockdown were not significantly associated with strength of the human-animal bond (in animal owners), but only with animal ownership in more general terms. Previous research has also pointed in this direction [30], and we believe our results provides momentum to the growing need for further rigorous research using appropriately nuanced instruments and variables in the area of human-animal interaction research.

Importantly, our study did not identify significant associations between regular interactions with non-companion animals and mental health and loneliness outcomes in non-animal owners. While we are not aware of previous studies that have investigated this link, we considered evidence of the potential benefits of human-animal interaction in more general terms [48] and hypothesized that engaging with non-domestic animals, for example through regular observations and feeding, may result in a human-animal relationship that could translate into benefits during the lockdown situation. Our bespoke instrument to measure this kind of human-animal interaction was not validated and overall perhaps crude, measuring presence of regular interaction and sustained frequency since lockdown. It is possible that we missed a potential effect–either due to instrument weakness, or due to a relatively small sub sample size of non-animal owners. An alternative explanation for our 'negative' finding is that our data in fact support the social buffering hypothesis in that some form of clear social affiliation with the animal as a recognised member of the human's social group may need to exist for the buffering effect to occur. Future research could explore the potential role of human-animal interactions and relationships that form as a result of engagement with non-companion animals further. Investigating its potential impact on mental health and wellbeing, perhaps particularly in isolated or otherwise vulnerable populations (e.g. the elderly), for whom caring for a companion animal may not be a feasible prospect, appears timely and important.

In conclusion, our study demonstrated that human-animal bond measures may constitute indicators of mental health-related vulnerability in animal owners, and that the strength of the bond in terms of intimacy/closeness did not depend on animal species. It highlighted the role of companion animals as potential social buffers for psychological distress and loneliness, regardless of species. Further targeted investigations relating to these important areas of human health, including the development of nuanced instruments, are essential.

## Supporting information

**S1 Table. Complete participant characteristics.**
(DOCX)

**S2 Table. Mental health and loneliness scores for animal owners and non-owners, pre- and since lockdown.**
(DOCX)

## Author Contributions

**Conceptualization:** Elena Ratschen, Emily Shoesmith, Lion Shahab, Karine Silva, Paul Toner, Catherine Reeve, Daniel S. Mills.

**Data curation:** Emily Shoesmith, Lion Shahab.

**Formal analysis:** Emily Shoesmith, Lion Shahab, Dimitra Kale.

**Investigation:** Elena Ratschen, Emily Shoesmith, Lion Shahab, Dimitra Kale.

**Methodology:** Elena Ratschen, Emily Shoesmith, Lion Shahab, Dimitra Kale.

**Project administration:** Elena Ratschen, Emily Shoesmith, Lion Shahab.

**Resources:** Elena Ratschen, Emily Shoesmith, Lion Shahab.

**Software:** Emily Shoesmith, Lion Shahab.

**Supervision:** Elena Ratschen.

**Writing – original draft:** Elena Ratschen, Emily Shoesmith, Lion Shahab, Karine Silva, Dimitra Kale, Paul Toner, Catherine Reeve, Daniel S. Mills.

**Writing – review & editing:** Elena Ratschen, Emily Shoesmith, Lion Shahab, Karine Silva, Dimitra Kale, Paul Toner, Catherine Reeve, Daniel S. Mills.

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
