## [Decision Letter · Decision Letter 0]

27 Aug 2020

PONE-D-20-22780

Human-animal relationships and interactions during the Covid-19 lockdown phase in the UK: investigating links with mental health and loneliness.

PLOS ONE

Dear Dr. Ratschen,

Thank you for submitting your manuscript to PLOS ONE. After careful consideration, we feel that it has merit but does not fully meet PLOS ONE’s publication criteria as it currently stands. Therefore, we invite you to submit a revised version of the manuscript that addresses the points raised during the review process.

Three Reviewers have evaluated the manuscript, providing generally favourable opinions. I encourage Authors to address Reviewers' concerns, paying particular attention to details in methods and results, and to indications for future research and the possible evolution of the health emergency scenario.

We look forward to receiving your revised manuscript.

Kind regards,

Stefano Triberti, Ph.D.

Academic Editor

PLOS ONE

Journal Requirements:

Reviewers' comments:

Reviewer's Responses to Questions

**Comments to the Author**

1. Is the manuscript technically sound, and do the data support the conclusions?

Reviewer #1: Yes

Reviewer #2: Yes

Reviewer #3: Partly

2. Has the statistical analysis been performed appropriately and rigorously? 

Reviewer #1: Yes

Reviewer #2: Yes

Reviewer #3: Yes

3. Have the authors made all data underlying the findings in their manuscript fully available?

Reviewer #1: Yes

Reviewer #2: Yes

Reviewer #3: Yes

4. Is the manuscript presented in an intelligible fashion and written in standard English?

Reviewer #1: Yes

Reviewer #2: Yes

Reviewer #3: No

5. Review Comments to the Author

Reviewer #1: The paper by Shoesmith and colleagues describes a research aimed at investigating human-animal relationship during the lockdown due to covid-19 emergency. The authors prepared a detailed questionnaire that was submitted to more than 5.000 participants. Results highlighted that companion animals were perceived as an important source of support. However, poorer mental health pre-lockdown was associated with a stronger reported human-animal bond, even if animal ownership was associated with smaller decreases in mental health and smaller increases in loneliness since lockdown. The manuscript is well written and complete. The topic is timely and interesting. I believe it can be considered for publication in Plosone, even if I would have appreciated some more details about if and how the relationship was changed during quarantine (e.g. the way to look at the animal, the quality of the time spent together, if this period has strengthened their bond…).

I have just few minor issues in mind that the authors could address:

- I have some concerns about how these two questions are formulated: ‘my animal has positive effects on my family at this time’; ‘my animal causes problems in my family at this time’. I was wondering why the authors chose to use the term “family” which excludes for example people living alone or with other non-relatives.

- The authors report a questionnaire section which included nine statements (yes/no) relating to practical aspects of animal ownership impacted by the pandemic. Can the authors also briefly report the missing items here (that are anyhow reported at page 12?

Reviewer #2: This study was fascinating. I'm impressed by the multiple angles with which the authors explored the results and their interpretations. I would have liked to have seen a greater description of implications for the continued pandemic and/or should a similar event occur in the future. Regardless, even without more details around this topic, I find this to be an article worthy of publication.

Reviewer #3: The study reports results from an online survey investigating the human-animal bond and human-animal interactions and its relationship with mental health and loneliness as well as the role of the animals during the Covid-19 lockdown in UK.

Major comments

This study targets an innovative aspect within the field of human-animal interaction and looks at important questions. The authors carefully discuss limitations that are clearly given as the sample is not representative and consists of mostly “animal lovers”. However, the conclusions are too general and I think it is an overstatement to conclude that “the study demonstrated the potential of human-animal bond measures as adjunct screening instruments for or indicators of mental health-related vulnerability in animal owners.” (lines 581 – 583). There are still too many open questions to use this measure in clinical context, however this is a relevant finding that should be replicated in representative datasets (within animal owners, and in a next needed step also in the general population). Moreover, this statement reflects only RQ3. The fact that animal species was not as relevant for emotional closeness, for example, seems equally important for our understanding of human-animal interactions.

The study is nicely structured but the flow of the text should be improved. Often, the sentences are long and the wording is complex. Moreover, the authors put a lot of text within brackets what makes the text hard to read and not fluent. Usually, it is not necessary to put the text in brackets. If the information is relevant, it can be introduced within the text as normal sentence, e.g. line 182/183 or 186.

It is not clearly defined what the authors mean by companion respective non-companion animals. Please define this in the introduction or the methods section. Are horses, for example, companion or non-companion animals? Both companion and non-companion animals might be owned or not (except for wildlife that is not owned). How was this aspect taken into account?

Minor comments:

Recruitment and Procedures: Was it clear that the questionnaire was about human-animal interaction? Please give a bit more context of how the questionnaire was designed and advertised.

Measuring companion animal ownership (line 161): how did the authors make sure that the participants knew what an emotional support animal, an assistance or a therapy animal is? This is something that is not always clear to the public. Please elaborate on this and how the questionnaire looked. It might be helpful to have it in the supplementary materials.

Remote social contact during Covid-19: Were there only three answer possibilities or are these examples? Currently, it is just indicated that they range from ‘many times per day’ over ‘less than once per week’ to ‘I do not have family members or friends I am in regular contact with’.

Data analysis: The authors should again state their questions and not just put the abbreviations such as RQ1 there or in brackets. It is not very reader friendly in this way.

It is a strong assumption that a construct like human-animal bond is a continuous variable. Please justify this or discuss how this affected interpretation of the results.

There are several small inconsistencies in formatting (d vs d or thousand with or without comma within a number e.g. line 283)

Figure 1: Do the error bars indicate SD or SEM? Please indicate.

Figure 2: The y-axis is not labelled.

6. PLOS authors have the option to publish the peer review history of their article (what does this mean?). If published, this will include your full peer review and any attached files.

Reviewer #1: No

Reviewer #2: No

Reviewer #3: No

---

## [Author Response · Author response to Decision Letter 0]

4 Sep 2020

We thank the reviewers for their constructive and helpful comments, which we address point by point below. In line with requests from the Editor, we have revised our manuscript ‘paying particular attention to details in methods and results, and to indications for future research and the possible evolution of the health emergency scenario’.

We are grateful for the opportunity to submit a revised version of our manuscript and hope that we have addressed all matters raised satisfactorily. 

Reviewer One:

1. I have some concerns about how these two questions are formulated: ‘my animal has positive effects on my family at this time’; ‘my animal causes problems in my family at this time’. I was wondering why the authors chose to use the term “family” which excludes for example people living alone or with other non-relatives.

These questions were aimed at circumstances in which a companion animal, or the Covid-19-related circumstances interfering with its care or other aspects of ownership, may have been perceived as beneficial or challenging for the family context. The term family does not typically exclude non-relatives (e.g. married couples or partners) and for people living alone and without family, i.e. those who felt that the questions did not apply to them, a response option of ‘Don’t know’ was available to answer the questions. Importantly, there are no missing data for these questions, indicating that respondents found the questions clear. 

2. The authors report a questionnaire section which included nine statements (yes/no) relating to practical aspects of animal ownership impacted by the pandemic. Can the authors also briefly report the missing items here (that are anyhow reported at page 12?)

All nine statements are now listed in the methods section under measures (see p.8 of the revised manuscript).

Reviewer Two:

This study was fascinating. I'm impressed by the multiple angles with which the authors explored the results and their interpretations. I would have liked to have seen a greater description of implications for the continued pandemic and/or should a similar event occur in the future. Regardless, even without more details around this topic, I find this to be an article worthy of publication.

We thank reviewer 2 for their positive feedback and have included some content in the discussion to highlight implications for the continued pandemic (p.23-24), as far as this was reasonable within the remit of our study. 

Reviewer Three:

1. The conclusions are too general, and I think it is an overstatement to conclude that “the study demonstrated the potential of human-animal bond measures as adjunct screening instruments for or indicators of mental health-related vulnerability in animal owners.” (lines 581 – 583). There are still too many open questions to use this measure in clinical context, however this is a relevant finding that should be replicated in representative datasets (within animal owners, and in a next needed step also in the general population). Moreover, this statement reflects only RQ3. The fact that animal species was not as relevant for emotional closeness, for example, seems equally important for our understanding of human-animal interactions.

We have revised our conclusions in the abstract and main manuscript to reflect the content of these helpful comments (p.2 and p.24) and have also attenuated statements relating to the potential of human-animal bond measures as adjunct screening instruments in the discussion section (p.21). 

2. The study is nicely structured, but the flow of the text should be improved. Often, the sentences are long, and the wording is complex. Moreover, the authors put a lot of text within brackets what makes the text hard to read and not fluent. Usually, it is not necessary to put the text in brackets. If the information is relevant, it can be introduced within the text as normal sentence, e.g. line 182/183 or 186.

As suggested we have revised the manuscript throughout to reduce the complexity of sentences and removed brackets and excessive paraphrasing. 

3. It is not clearly defined what the authors mean by companion respective non-companion animals. Please define this in the introduction or the methods section. Are horses, for example, companion or non-companion animals? Both companion and non-companion animals might be owned or not (except for wildlife that is not owned). How was this aspect taken into account?

We defined companion animals in accordance with the British Small Animals Veterinary Association (BSAVA), which states: ‘a companion animal is any domestic-bred or wild-caught animals, permanently living in a community and kept by people for company, enjoyment, work (e.g. support for blind or deaf people, police or military dogs) or psychological support – including, but not limited to dogs, cats, horses, rabbits, ferrets, guinea pigs, reptiles, birds and ornamental fish’. Horses are therefore included. 

The original question posed to survey respondents reads as follows: ‘Do you have any animals that live with you or near you, and that you or anyone in your household are the main caretaker of? Please do not include animals kept as livestock (e.g. farm sheep or cattle) in your response.’

We have now included this information in the methods section under measures (see p.6 of the revised manuscript). 

4. Recruitment and Procedures: Was it clear that the questionnaire was about human-animal interaction? Please give a bit more context of how the questionnaire was designed and advertised.

We have now added information to the methods section under Recruitment and Procedures to clarify that the Participant Information sheet included an overview of the study and its aim to investigate the role human-animal interactions and relationships play for health and wellbeing in the context of the Covid-19 pandemic (p.9). 

We have included more information about how the questionnaire was designed in the Measures subsection (p.5):

‘A bespoke questionnaire was generated to enable collection of data most relevant to answer the study questions. It was developed by a multi-disciplinary team of academics with input from third sector animal welfare and training organisations. The questionnaire included validated items and new items based on expert consensus relating to emerging Covid-19-related aspects, as detailed below’.

5. Measuring companion animal ownership (line 161): how did the authors make sure that the participants knew what an emotional support animal, an assistance or a therapy animal is? This is something that is not always clear to the public. Please elaborate on this and how the questionnaire looked. It might be helpful to have it in the supplementary materials.

We agree that the definitions in question may not be fully understood by the public in general, but we believe that people who have an assistance, therapy or emotional support animal, i.e. the subgroup of respondents this question was primarily aimed at, are aware of them. People who live with a specialist trained assistance dog, people who provide animal-assisted interventions/therapy with their animal, or people whose animal is an emotional support animal (https://www.emotionalsupportanimals.org.uk/uk-law/) will know that they live with such an animal. Others had the option to respond ‘none of the above’ to the question. There are no missing data for this question and no indication that respondents struggled to understand the question. 

We have included further detail on how the question looked in the manuscript and elaborated on the answer options as suggested (p.6). 

6. Remote social contact during Covid-19: Were there only three answer possibilities or are these examples? Currently, it is just indicated that they range from ‘many times per day’ over ‘less than once per week’ to ‘I do not have family members or friends I am in regular contact with’.

All of the possible responses to the remote social contact question are now listed in the methods section under measures (see p.8 of the revised manuscript). 

7. Data analysis: The authors should again state their questions and not just put the abbreviations such as RQ1 there or in brackets. It is not very reader friendly in this way.

We have now stated the specific research questions again in the Data Analysis section as suggested. 

8. It is a strong assumption that a construct like human-animal bond is a continuous variable. Please justify this or discuss how this affected interpretation of the results.

As detailed in the methods section, our human-animal bond variable is based on the validated Comfort from Companion Animals (CCA) construct by Zasloff (1996), which results in a score. We are not aware of any literature in which this score would have been categorised/treated as a non-continuous variable. We have followed the example of existing studies, in which the score was treated as a continuous variable (Blazina & Kogan, 2019; Le Roux & Wright, 2020; Luhmann & Kalitzi, 2018). It is recognised that ‘while Likert questions or items may well be ordinal, Likert scales, consisting of sums across many items, will be interval. It is completely analogous to

the everyday, and perfectly defensible, practice of treating the sum of correct answers on a multiple choice test, each of which is binary, as an interval scale.’ (Norman, 2010) 

9. There are several small inconsistencies in formatting (d vs d or thousand with or without comma within a number e.g. line 283)

All thousands within a number now have commas. We have edited for consistency with cursive vs plain formats of p in the text where p-values are reported. 

10. Figure 1: Do the error bars indicate SD or SEM? Please indicate.

We believe this comment was intended for Figure 2. Figure 2 now indicates that the error bars represent SEM in the Figure caption.

11. Figure 2: The y-axis is not labelled.

We believe this comment was intended for Figure 1. We have now labelled the y-axis on Figure 1. Thank you. 

References

Marieanna C. le Roux & Simonne Wright (2020) The Relationship Between Pet Attachment, Life Satisfaction, and Perceived Stress: Results from a South African Online Survey, Anthrozoös, 33:3, 371-385, DOI: 10.1080/08927936.2020.1746525

Chris Blazina & Lori Kogan (2019) Do Men Underreport and Mask Their Emotional Attachment to Animal Companions? The Influence of Precarious Masculinity on Men’s Bonds with Their Dogs, Anthrozoös, 32:1, 51-64, DOI: 10.1080/08927936.2019.1550281

Maike Luhmann & Anna Kalitzki (2018) How animals contribute to subjective well-being: A comprehensive model of protective and risk factors, The Journal of Positive Psychology, 13:2, 200-214, DOI: 10.1080/17439760.2016.1257054

Norman G. Likert scales, levels of measurement and the "laws" of statistics. Adv Health Sci Educ Theory Pract. 2010;15(5):625-632. doi:10.1007/s10459-010-9222-y

Zasloff RL. (1996). Measuring attachment to companion animals: a dog is not a cat is not a bird. Applied Animal Behaviour Science, 47:43-48.

---

## [Editor Report · Decision Letter 1]

7 Sep 2020

Human-animal relationships and interactions during the Covid-19 lockdown phase in the UK: investigating links with mental health and loneliness.

PONE-D-20-22780R1

Dear Dr. Ratschen,

We’re pleased to inform you that your manuscript has been judged scientifically suitable for publication and will be formally accepted for publication once it meets all outstanding technical requirements.

Kind regards,

Stefano Triberti, Ph.D.

Academic Editor

PLOS ONE
---

## [Editor Report · Acceptance letter]

18 Sep 2020

PONE-D-20-22780R1 

Human-animal relationships and interactions during the Covid-19 lockdown phase in the UK: investigating links with mental health and loneliness 

Dear Dr. Ratschen:

I'm pleased to inform you that your manuscript has been deemed suitable for publication in PLOS ONE. Congratulations! Your manuscript is now with our production department. 

Kind regards, 

on behalf of

Dr. Stefano Triberti 

Academic Editor

PLOS ONE